# Combination Therapy for the Treatment of Shingles with an Immunostimulatory Vaccine Virus and Acyclovir

**DOI:** 10.3390/ph16020226

**Published:** 2023-02-01

**Authors:** Tibor Bakacs, Volker Sandig, Imre Kovesdi

**Affiliations:** 1Department of Probability, Alfred Renyi Institute of Mathematics, The Eotvos Lorand Research Network (ELKH), 1053 Budapest, Hungary; 2ProBioGen AG, 13086 Berlin, Germany; 3Unleash Immuno Oncolytics Inc., St. Louis, MO 63110, USA

**Keywords:** herpes, shingles, herpes zoster ophtalmicus, acyclovir, infectious bursal disease virus, IBDV, adjunct therapy, superinfection, interferons, varicella zoster virus, VZV

## Abstract

Practically the entire global population is infected by herpesviruses that establish lifelong latency and can be reactivated. Alpha-herpesviruses, herpes simplex viruses 1 and 2 (HSV-1/HSV-2) and varicella zoster virus (VZV), establish latency in sensory neurons and then reactivate to infect epithelial cells in the mucosa or skin, resulting in a vesicular rash. Licensed antivirals inhibit virus replication, but do not affect latency. On reactivation, VZV causes herpes zoster, also known as shingles. The 76-year-old first author of this paper published an autobiography of his own severe herpes zoster ophthalmicus (HZO) infection with orbital edema, which is considered an emergency condition. Acyclovir (ACV) treatment was complemented with an immunostimulatory viral therapy, which resolved most symptoms within a few days. The orally administered live-attenuated infectious bursal disease vaccine virus (IBDV) delivers its double-stranded RNA (dsRNA) cargo to host cells and activates the natural antiviral interferon (IFN) gene defense system from within the host cells. IBDV has already been demonstrated to be safe and effective against five different families of viruses, hepatitis A virus (HAV), hepatitis B and C virus (HBV/HCV), severe acute respiratory syndrome coronavirus 2 (SARS-CoV-2), and varicella zoster virus (VZV). Here we propose a short phase I/II trial in elderly shingles patients who will be assigned to receive either ACV monotherapy or ACV combined with R903/78, an attenuated immunostimulatory IBDV strain. The primary endpoints will be safety, but the efficacy of the combination therapy against the ACV monotherapy also will be assessed.

## 1. Introduction

Practically all humans are infected with herpesviruses, which establish a lifelong latency that can be reactivated. Alpha-herpesviruses, herpes simplex viruses 1 and 2 (HSV-1 and HSV-2) and varicella zoster virus (VZV), establish latency in sensory neurons and reactivate from neurons to infect epithelial cells in the mucosa or skin, resulting in a vesicular rash [1]. Antiviral drugs inhibit virus replication, but they have no effect on latent viral DNA, which is stably maintained in the nucleus of cells as multiple copies of circular episomes.

VZV is neurotropic and infects nearly all humans [2]. The primary infection causes chickenpox (varicella), when vesicles can develop on any dermatome. In such a way cell-free VZV has direct access to dorsal root, cranial nerve, and autonomic nervous system ganglia, where it becomes latent. Since VZV-specific host immunity declines with age, the virus may reactivate anywhere on the body. When VZV is reactivated, it causes herpes zoster (HZ or shingles). VZV travels from the cell bodies of the neurons to the nerve terminals in the skin. This process is characterized by the dermatomal distribution of pain, rash and infectious vesicles. Healing of shingles is completed in two to four weeks. Post-herpetic neuralgia (PHN) is, however, a frequent complication, where pain persists for months or years. About one-third of people are affected by shingles during their lifetime, while one-half of those who live to 80 years are affected. Shingles incurs significant costs worldwide. Widespread vaccine implementation on a global scale is hindered by multiple barriers [3].

When VZV reactivates in the cranial nerve (CN) V, herpes zoster ophthalmicus (HZO) develops [4]. HZO accounts for 10–15% of HZ cases. HZO is characterized by vesicular and erythematous involvement of the CN V1 dermatome. If orbital edema develops the patient must be referred immediately for specialized evaluation and treatment, as this condition is an ophthalmologic emergency.

Recently, the 76-year-old first author of this paper suffered from a severe HZO infection that was accompanied by orbital edema [5]. An intermittent throbbing pain was felt in more than three dermatomes including the frontal, orbital, temporal, and occipital/nuchal areas. ACV treatment was administered belatedly, because the first symptoms were atypical. Conventional treatment was therefore complemented with an experimental immunostimulatory therapy using an attenuated veterinary vaccine virus, IBDV (Figure 1A) [6]. Most symptoms resolved within a few days. Given the author’s advanced age, and the belated administration of conventional ACV treatment, the significant acceleration of recovery is unlikely to be explained by the ACV treatment alone (Figure 1B).

In order to confirm and extend this case report, here we propose a small phase I/II clinical trial in which elderly patients with herpes zoster will receive either ACV monotherapy or ACV plus the experimental infectious bursal disease vaccine virus (IBDV), strain R903/78 as adjunct therapy. The primary endpoint will be safety, but efficacy of the combined therapy against the ACV monotherapy will be also evaluated.

## 2. Repurposing Is a Drug Development Strategy

Repurposing received heightened attention following the U.S. Food and Drug Administration granted emergency use authorization of several repurposed drugs to treat COVID-19. Repurposing drugs can be faster, cheaper, less risky and carry higher success rates than traditional drug development, primarily because it can bypass earlier stages of development. While de novo drug development can take 10–17 years, repurposed drugs are generally approved within 3–12 years, and at about half the cost [7]. In contrast, any modification will have consequences in terms of production costs, good manufacturing practices, and regulatory requirements to conduct clinical trials [8]. Therefore, use of vaccines ‘as is’ may represent a cost-effective strategy to bring new therapeutic options to patients. Repurposing, however, requires special care in terms of clinical development and regulatory requirements.

Vandeborne et al., identified repurposing opportunities in the use of 16 infectious disease vaccines to stimulate immune responses to cancer, focusing on the use of existing licensed vaccines as add-ons to registered immunotherapies [9].

A somewhat similar strategy based on viral interference was recently proposed for the therapeutic control of non-related viral diseases by utilizing naturally or intentionally attenuated viruses, which are harmless to people [6].

Combining the two above strategies, live attenuated vaccines could be used for the immediate mitigation and control of unexpected pandemic outbreaks before prophylactic vaccines and/or pathogen-specific therapies are developed [10]. The proof of principle of this idea was demonstrated using a bivalent oral poliovirus vaccine (bOPV), which significantly reduced the number of COVID-19 cases compared to placebo [11].

IBDV have been used as a chicken vaccine for decades without causing any known human disease. In the viral interference context, IBDV has already been demonstrated to be safe and effective against five different families of viruses (see Table 1 for references in [12]). Therefore, the use of the attenuated IBDV-R903/78 drug candidate for the treatment of VZV caused disease might be a very appealing proposal.

## 3. The Rationale of Using Repurposed IBDV Veterinary Vaccine to Control Unrelated Viral Infections

The presence of viruses is seen by host cells via the so-called pattern recognition receptors (PRRs) which sense viral nucleic acids by recognizing pathogen-associated molecular patterns (PAMPs). PRRs elicit transcriptional activation and mediate the initiation of antigen-specific adaptive immune responses and release of inflammatory cytokines. Double-stranded (ds)RNA is a molecular pattern which is produced by most viruses at some point during their replication cycle. Toll-like receptors (TLRs) are PRRs that recognize dsRNA [13]. Type I interferons (IFNs) are induced by dsRNA, which then triggers the intracellular IFN signaling pathway that activates the IFN antiviral gene defense system [14].

Viruses can concurrently or sequentially infect their target cells leading to virus–virus interactions. Infection by a first virus could inhibit replication of a second virus by IFN induction, which is called viral interference. As the IFN response could confer a temporary nonspecific immunity to the host, it can control most, if not all, virus infections before adaptive immunity develops. It has been proposed that viral induction of a nonspecific localized temporary state of immunity may provide a strategy to control viral infections. This idea was translated into a safe and effective preventive and therapeutic strategy in the late 1960s. In large-scale clinical studies involving about 320,000 people in 16 regions of three republics of the former Soviet Union, IFN-inducing live enteroviral vaccine (LEV) strains provided temporary protection and treatment against seasonal influenza and acute respiratory infections [15].

### 3.1. The IBDV Superinfection Therapy (SIT) Is an Intentional Viral Coinfection Strategy

The conventional description of viral interference states that an established infection suppresses the replication of a coinfecting virus. In HBV- or HCV-infected patients, however, it was observed that the coinfecting virus could also be dominant over the replication of the initial infecting virus. While dominance can alternate between HBV and HCV viruses, typically HBV appears to be suppressed by HCV. Notwithstanding, when HBV is suppressed by HCV, hepatitis persists [16]. Natural coinfection can, however, be intentionally reproduced by coinfecting the patients with a harmless virus. Intentional coinfection of patients by a harmless virus is the clinically validated superinfection therapy (SIT). The first drug candidate of SIT was a conventionally produced attenuated avian vaccine virus, the IBDV, which is a potent activator of the IFN-dependent antiviral gene program. Superinfection is a host-directed therapy, during which the non-pathogenic avian live-attenuated IBDV vaccine delivers its dsRNA cargo to host cells, activating their natural antiviral IFN gene defense system from within [17]. IBDV induces one of the strongest interferon responses among viruses. The orally administered IBDV has already been effective against five different families of viruses, including hepatitis A virus (HAV), hepatitis B and C viruses (HBV, HCV), severe acute respiratory syndrome coronavirus 2 (SARS-CoV-2), and VZV (see Table 1 for references in [12]).

### 3.2. The Safety of the Veterinary IBDV Vaccine Drug Candidate

For any virus, the ability to jump between hosts depends on the evolutionary relatedness of the donor and recipient species. The natural hosts of IBDV are the domestic fowl, including chickens and turkeys [17]. As birds and mammals diverged more than 200 million years ago, thirteen mutations would be required for avian influenza viruses to establish productive infections in humans [18]. For such a gigantic jump, the influenza virus requires an intermediate host (the swine) in order to pre-adapt to humans. IBDV, fortunately, does not have such a natural intermediate host. Not unexpectedly, no zoonosis cases were ever reported during IBDV mass-vaccination programs in poultry over the past half a century. However, even a very low risk of zoonosis is a legitimate regulatory concern. Therefore, reverse genetics technology was used to create the IBDV-R903/78 drug candidate, which provided a homogenous starting material to ensure batch-to-batch consistency [19].

In stark contrast to the systemic toxic interferon-α-2b (IFNα2b) therapy, which may require dose modification or discontinuation, only minimal flu-like side effects were observed during IBDV superinfection therapy, even in parenchymally decompensated severe hepatitis patients [20,21,22,23]. One of the reasons for this difference could be that IFN receptors are virtually ubiquitously expressed in all cell membranes, and, following the interaction of IBDV with host cells, its dsRNA is recognized by specific receptors, which activate several gene families from within the cells.

## 4. IBDV R903/78 Drug Candidate Is Simple to Manufacture and Will Be Affordable in Resource-Limited Countries

For manufacturing of the IBDV-R903/78 drug candidate an immortalized embryonic duck cell line, AGE1.CR.PIX, was chosen for its exceptionally high viral productivity. AGE1.CR.PIX was adapted to propagate in defined solutions and media. This cell line had already been used for the production of clinical grade live attenuated vectored vaccines based on Modified Vaccinia virus Ankara (MVA) [24]. AGE1.CR.PIX has been selected for its high permissivity and titers exceeding other cell lines by 100–1000-fold, reaching levels between 10^9^ and 10^10^ infectious units (IU)/mL. Assuming a typical clinical dose of 10^7^ IU, 100,000 to 1,000,000 doses can be produced in 1L of fermentation volume. For the manufacture of parenteral live vaccines, a suitable purification strategy is applied to reduce host cell protein contamination and to assure levels of host cell DNA below 10 ng/dose. The oral application proposed in this clinical trial for IBDV-R903/78 requires less stringent reductions of host cell DNA and protein. The manufacturing process is furthermore simplified, as IBDV-R903/78 is secreted into the cell culture supernatants not requiring lysing of the cells, greatly reducing downstream purification complexity and expense. The expected high virus concentration in the yield combined with oral delivery allows for a simple formulation methodology.

## 5. Phase I/II Combination Clinical Trial of ACV Plus IBDV-R903/78 Drug Candidate for the Treatment of Facial HZ in Elderly Patients

Case studies can generate hypothesis and provide guidance among populations with limited numbers of patients. Compassionate use or N-of-1 trials provide a mechanism for making evidence-based treatment decisions for an individual patient in the real-world clinical practice. N-of-1 trials use key methodological elements of group clinical trials to evaluate treatment effectiveness in a single patient, for example, in patients using concurrent therapies [25,26]. While the autobiographic HZO case study of the first author [5] was not designed as an N-of-1 trial, with the benefit of hindsight, several of its features would have suited for such trial. For example, the combination of ACV and IBDV therapy had quick onset and offset with a very short treatment period, making it reasonably easy to measure the outcomes of interest. Simple visual inspection alone produced accurate conclusions in order to determine the results of the trial. Based on this case study, we generated a hypothesis that could provide guidance for HZ patients. We predict that bolstering of innate antiviral immunity of the facial HZ and HZO patients by the administration of the IFN-inducing IBDV would leverage the activity of ACV.

Based on the unique HZO case study [5], here we propose a small randomized phase I/II trial of IBDV-R903/78 in elderly subjects with facial HZ in which we will assess safety and tolerability as primary endpoints, and immunogenicity and therapeutic efficacy as exploratory endpoints. We would like to confirm the previously demonstrated safety of IBDV-R903/78 in humans (see in [12]) in order to indicate a potential for use of IBDV-R903/78 as a combination therapy to improve the efficacy of ACV in facial HZ symptomatic subjects. We will compare the healing period of HZ symptoms using ACV monotherapy to that using the combination of ACV plus IBDV-R903/78 therapy as described below.

### 5.1. Design Overview

Patients with facial HZ will be enrolled in the study. Informed Consent from all study participants will be obtained. All patients will receive oral anti-viral medication (ACV) as a standard-of-care therapy. This will be a single center study to investigate the safety and efficacy of ACV compared to ACV and IBDV-R903/78 as adjunct therapy to determine whether the two regimens are equally effective for the treatment of facial HZ patients over a period of 7 days. Patients will be followed for 28 days, with intermediate visits on days 7 and 14.

The study will consist of two parts: (A) a lead-in dose escalation study of ACV combined with increasing concentration of IBDV-R903/78; (B) a comparison study of the ACV cohort with ACV combined with IBDV-R903/78 dose determined in part (A).

### 5.2. Study Objectives

#### 5.2.1. Primary

To determine the Maximum Tolerated Dose (MTD) of IBDV-R903/78, an attenuated vaccine strain following oral administration to subjects with facial HZ.

To determine the systemic toxicity of IBDV-R903/78 administered by oral administration.

#### 5.2.2. Secondary

To determine the biological effects and extent of IBDV-R903/78 as an adjuvant therapy for facial HZ, including the analysis of the T cell repertoire as well as the inflammatory cytokine marker profile.

### 5.3. Participants

#### 5.3.1. Inclusion Criteria

The inclusion criteria require immunocompetent adults older than 60 years with facial HZ and with a score of 40 mm or more on the visual analog scale (VAS) for at least two of the following symptoms/signs: pain, loss of sensitivity, burning, and pruritus for up to seven days.

#### 5.3.2. Exclusion Criteria

The exclusion criteria include having a history of hypersensitivity to acyclovir, previous use of antiviral drugs or corticosteroid therapy, and severe systemic disease; patients who received shingles vaccination will be excluded.

### 5.4. Randomization and Intervention

(A)Lead-in dose escalation study section

The starting dose of IBDV-R903/78 is supported by GLP toxicity studies. The autobiographic HZO case report indicated that seven days ACV and four days 1.0 × 10^6^ Infection Unit (IU) IBDV treatment was sufficient in a severe HZO infection [5]. All therapy will be given orally for 7 days. All patients will receive ACV 800 mg (two capsules) five times daily plus IBDV-R903/78 agent orally. The IBDV-R903/78 drug candidate containing four different doses will be escalated in, accordingly, to a 3 + 3 protocol until MTD is reached. The starting dose will be 1 log below the HZO study [5] dose, 1.0 × 10^5^ IU followed by 1.0 × 10^6^ IU, 1.0 × 10^7^ IU and 1.0 × 10^8^ IU maximum dose that will be administered orally, daily for 7 days.

(B)Comparative study section

Eligible patients will be randomly assigned according to a computer-generated randomization code (1:1) to either ACV 800 mg (two capsules) five times daily or ACV plus IBDV-R903/78 agent orally once a day, for 7 days. The daily IBDV-R903/78 IU dose will be determined by the MTD dose and efficacy, as assessed in part A of this section. For example, if there are no serious adverse events above grade 2 at the highest 1.0 × 10^8^ IU dose, then a 1 log lower dose will be used (1.0 × 10^7^ IU). However, the dose might be lowered further if a dose indicates efficacy at a lower level, for example at 1 × 10^6^ IU.

The number of subjects dosed will be 10 in each cohort. This number is determined by statistical evaluation. The average time to cure for ACV treated patients was ~15 days [27]. In the single patient HZO study [5], the time to cure was reduced to 5 days. To reach a significant level of 95% to be able to distinguish between the means of the two cohorts, a minimum of 10 subjects needs to be enrolled in each group.

### 5.5. Study Assessments

#### 5.5.1. MTD and Systemic Toxicity

Serious adverse events will be determined according to serum electrolytes, liver enzymes, creatinine, and the full blood count will be assessed at admission and before outpatient discharge. Decisions regarding discharge will be made on an individual basis after a substantial improvement in symptoms.

#### 5.5.2. Efficacy

After the baseline assessment, lesions will be evaluated by the investigator after 7 days of treatment and on days 7 and 21 post-therapy. The efficacy secondary endpoint is defined as the time to full crusting of facial HZ lesions. Secondary endpoints will be also determined as the proportion of patients who achieved complete cure and the individual score change for signs/symptoms (pain, vesicular lesions, loss of sensitivity, burning pain, and pruritus) according to the patient’s self-reported diary. Each sign/symptom will be evaluated once daily for 28 days based on a visual analog scale (VAS). Each patient will self-administer the questionnaire, filling out the form before the beginning of treatment and then every day thereafter, on the basis of symptoms in the previous 24 h.

#### 5.5.3. Safety

Serum electrolytes, liver enzymes, creatinine, and the full blood count will be assessed at admission and before outpatient discharge. Decisions regarding discharge will be made on an individual basis after a substantial improvement in symptoms.

#### 5.5.4. Statistical Methods

For all efficacy endpoints, the statistical analyses will be performed in the per-protocol (PP) population, which will include patients compliant with the study protocol. The sample size will be estimated to detect the difference between the mean time to full crusting of herpes zoster lesions in the ACV monotherapy and ACV plus IBDV-R903/78 combined therapy groups. Assuming a dropout rate of 10% after randomization, 12 randomized patients per group (24 total) will provide 95% power to show whether the mean time to full crusting with ACV is longer than that of ACV plus R903/78 at the 5% significance level.

## 6. Discussion

Viruses encode mechanisms to counteract the host’s response and support efficient viral replication. In such a way, they are able to minimize the antiviral power of IFNs [14]. For example, VZV has co-evolved with its human host for millions of years, during which VZV obtained various capabilities to evade the hosts’ immune responses [28]. VZV produces multiple open reading frames (ORFs) such as ORF12, ORF63, and ORF66, which inhibit apoptosis in cells critical for viral dissemination and the establishment of lifelong latency. VZV also interferes with the type 1 IFN pathway and the production of pro-inflammatory cytokines by the inhibition of interferon regulatory transcription factor 3 (IRF3) and the nuclear factor kappa light chain enhancer of activated B cells (NF-κB). In such a way, VZV modulates components of the intrinsic, innate and adaptive immune response in order to ensure viral dissemination and life-long latency.

While virome interactions with the host are still encompassed by a prevailing view of viruses as a source of deadly pathogens, the truth is that mutually beneficial coinfections are the rule, rather than the exception [29]. In such a way, combinations of virus and host genes can contribute to host fitness. Therefore, to counteract the immune inhibitory capabilities of VZV with the harmless IBDV, makes sense. This hypothesis was recently confirmed by the unexpectedly fast recovery of the first author from his severe HZO infection by using a combination of conventional ACV and immunostimulatory IBDV therapy [5].

The nucleoside analogue ACV, which was discovered in the early 1970s, is safe, well-tolerated and still the most commonly used drug in cases of HSV and VZV infections [30]. However, the possibility of viral resistance remains. Furthermore, there are still no drugs, even in clinical trials, that would eliminate herpesviruses from an infected person. Despite the high prevalence of HSV and VZV infections, vaccines and passive imunoprophylaxis are available only against VZV. These unresolved issues, in our view at least, would justify the development of a combination immunostimulatory IBDV therapy for HZ.

A major limitation of the proof-of-principle ACV combination IBDV therapy in HZO patients was that it was demonstrated only in an autobiographic case study. Although case reports lack statistical analyses, they provide clinical insights that may be missed in clinical trials [31]. For example, the first cord blood (CB) transplant, which was performed in 1988 in a patient with Fanconi anemia, paved the way for establishing CB banks [32]. To address this limitation, we are preparing a phase I/II clinical study as described above. More importantly, the proposed phase I/II trial of IBDV-R903/78 in facial HZ patients, which does not require quarantine, would pave the way for an innovative strategy to control other viral infections, including SARS-CoV-2.

## 7. Conclusions

The proposed short phase I/II study in patients with facial herpes zoster could confirm the observation obtained in a single HZO patient that the different efficacy of the ACV plus IBDV-R903/78 combination therapy and ACV monotherapy is clinically relevant. As approximately one out of every three people will suffer an episode of shingles during their lifetime, the new combination treatment could help many millions of patients with HZ infection.

## 8. Patents

Title of invention: Use of a Birnavirus for the treatment of a disease caused by varicella zoster virus; Filing Date: 18-Nov-2022; Serial No.: 18/056,924.

## Figures and Tables

**Figure 1 pharmaceuticals-16-00226-f001:**
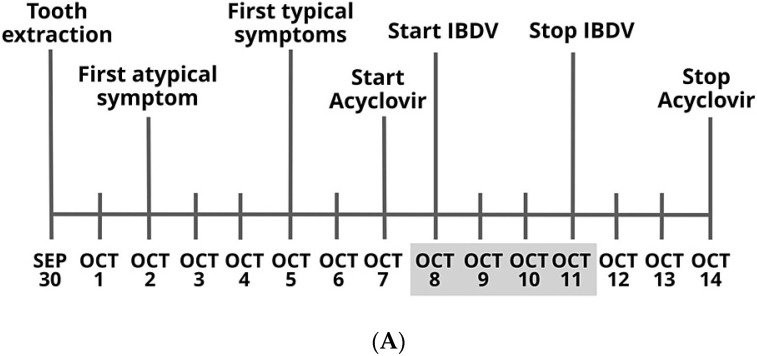
(**A**) Disease and treatment course: dateline of herpes zoster ophthalmicus and its treatment from 30 September to 14 October 2021. Extended lines represent the start of symptoms and treatment. Days highlighted in gray illustrate treatment with IBDV. (**B**) The author’s HZO with orbital edema at the peak of disease and in recovery. The “selfie” pictures were taken between 9 October 2021 and 12 October 2021. Consent to the publication of patient information was granted by Tibor Bakacs, M.D., Ph.D., D.Sc., as he was the patient and the treating physician in this autobiography. Reproduced from Bakacs, T. Healing of Severe Herpes Zoster Ophthalmicus Within a Few Days: An Autobiographical Case Report. *Cureus*. 2021, 13:e20303. doi:10.7759/cureus.20303 [5] with permission from Cureus Inc. (see in 8. Site Content and User Submissions; https://www.cureus.com/terms; accessed on 29 January 2023).

## Data Availability

Data is contained within the article.

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
