# Peer review of "Combination Therapy for the Treatment of Shingles with an Immunostimulatory Vaccine Virus and Acyclovir"

_pharmaceuticals, 2023, doi:10.3390/ph16020226_

Round 1

Reviewer 1 Report

The article submitted Bakacs et al to Pharmaceuticals is seeking the opinion for the proposed short phase I/II study in patients with facial herpes zoster with a combination therapy with orally administered live-attenuated infectious bursal diarrhea virus (IBDV; R903/78 strain) and acyclovir (ACV) in comparison to the monotherapy with ACV only. The proposed study is an outcome of the observation obtained in a single HZO patient who is the first author of this study. I appreciate the effort taken by the authors to translate the observation into a clinical relevant study.

My opinions for this study are listed below

(i) Using IBDV as a superinfection therapy candidate is a wonderful approach as it is attenuated and is a pathogen for an evolutionary diverged host. Hence the safety of the candidate is not a concern

(ii) For the study assessments, apart from the MTD and systemic toxicity I would recommend the authors to have a look at the T cell repertoire as well as inflammatory cytokine marker profile, which will give an additional insight in how the combinatorial as well as monotherapy acts in each case.

(iii)The method should mention getting the informed consent from the patients.

Author Response

We are grateful to Reviewer No.1. for his/her supporting opinion and important comments, which were incorporated into the text as follows.

1) A statement concerning the analysis of the T cell repertoire as well as inflammatory cytokine marker profile was inserted in line 231.

2) A request for Informed Consent from all study participants was inserted in line 213.

Reviewer 2 Report

Thank you for the opportunity to review this interesting manuscript about combination therapy for shingles. This study highlights the importance of better treatments for VZV. 

Please consider the following comments:

Page 3, line 101: Typing error, in the text is written prof instead of proof

Page 3, line 105: I suggest modifying from "IBDV have been used as a chicken vaccine for decades without causing any human disease" to "IBDV have been used as a chicken vaccine for decades without causing any known human disease"

Page 3, line 106 and page 4, line 148: In both sentences the authors comment that IBDV is effective against five different virus families but the sources are not cited in the text.

Page 6, line 235:I suggest that the authors clarify up to how many days after the onset of symptoms patients will be candidates for the therapeutic maneuver.

Page 6, line 237: Please describe whether to include or exclude patients who received Shingles Vaccination

Author Response

We are grateful to Reviewer No.2., for his/her encouraging opinion; the requested corrections were incorporated into the text as follows.

1) Misspelling of <proof> was corrected in line 102.

2) The word <known> was inserted in line 105.

3) The sources of the references were given as follows <see Table 1 for references in [12]> in lines 107 and 149, respectively.

4) The length of the treatment was specified in line 237.

5) Exclusion criteria were completed by this statement <patients who received shingles vaccination will be excluded> in line 240.